# Environmentally Friendly New Catalyst Using Waste Alkaline Solution from Aluminum Production for the Synthesis of Biodiesel in Aqueous Medium

**DOI:** 10.3390/bioengineering10060692

**Published:** 2023-06-07

**Authors:** Sandro L. Barbosa, David Lee Nelson, Lucas Paconio, Moises Pedro, Wallans Torres Pio dos Santos, Alexandre P. Wentz, Fernando L. P. Pessoa, Foster A. Agblevor, Daniel A. Bortoleto, Maria B. de Freitas-Marques, Lucas D. Zanatta

**Affiliations:** 1Department of Pharmacy, Federal University of Jequitinhonha and Mucuri Valleys-UFVJM, Campus JK, Rodovia MGT 367–Km 583, nº 5.000, Alto da Jacuba, Diamantina 39100-000, Brazil; dleenelson@gmail.com (D.L.N.); lucas.paconio@ufvjm.edu.br (L.P.); moises.pedro@ufvjm.edu.br (M.P.); wallanst@ufvjm.edu.br (W.T.P.d.S.); wentzap@hotmail.com (A.P.W.); 2University Center SENAI-CIMATEC, Av. Orlando Gomes, 1845, Piatã, Salvador 41650-010, Brazil; fernando.pessoa@fieb.org.br; 3Utah Science Technology and Research (USTAR), Biological Engineering, Utah State University, Logan UT620 East 1600 North, Suite 130, Logan, UT 84341, USA; foster.agblevor@usu.edu; 4Department of Geosciences, Universidade Federal do Pará, R. Augusto Corrêa, 01–Guamá, Belém 66075-110, Brazil; dabortoleto@yahoo.com.br; 5Department of Chemistry, Instituto de Ciências Exatas, Universidade Federal de Minas Gerais, Av. Antônio Carlos, 6627, Pampulha, Belo Horizonte 31270-901, Brazil; betanialf@hotmail.com; 6Laboratório de Química Bioinorgânica, Departamento de Química, Faculdade de Filosofia, Ciências e Letras de Ribeirão Preto, Universidade de São Paulo, Av. Bandeirantes, 3900, Ribeirão Preto 14040-901, Brazil; lucaszanatta@alumni.usp.br

**Keywords:** environmentally friendly processes, bayer residue, waste management, basic catalyst, contaminants, red mud, fatty acid methyl ester

## Abstract

Red mud (RM) is composed of a waste alkaline solution (pH = 13.3) obtained from the production of alumina. It contains high concentrations of hematite (Fe_2_O_3_), goethite (FeOOH), gibbsite [Al(OH)_3_], a boehmite (AlOOH), anatase (Tetragonal–TiO_2_), rutile (Ditetragonal dipyramidal–TiO_2_), hydrogarnets [Ca_3_Al_2_(SiO_4_)_3−x_(OH)_4x_], quartz (SiO_2_), and perovskite (CaTiO_3_). It was shown to be an excellent catalytic mixture for biodiesel production. To demonstrate the value of RM, an environmentally friendly process of transesterification in aqueous medium using waste cooking oil (WCO), MeOH, and waste alkaline solution (WAS) obtained from aluminum production was proposed. Triglycerides of WCO reacted with MeOH at 60 °C to yield mixtures of fatty acid methyl esters (FAMEs) in the presence of 0.019% (*w*/*w*) WAS/WCO using the WAS (0.204 mol L^−1^, predetermined by potentiometric titration) from aluminum production by the Bayer process. The use of the new catalyst (WAS) resulted in a high yield of the products (greater than 99% yield).

## 1. Introduction

Waste alkaline solution (WAS) and red mud (RM) (bauxite residue, bauxite tailings, red sludge, or alumina refinery residues) are industrial wastes generated during the processing of bauxite into alumina using the Bayer process (over 95% of the alumina is produced globally through the Bayer process) [1,2,3,4,5]. With every ton of alumina produced, approximately 1 to 1.5 tons of RM is also produced. Annual production of alumina in 2020 was over 133 million tons, resulting in the generation of over 175 million tons of RM [6]. Because of this large production and the material’s high alkalinity (pH of 10.5–12.5) caused by an elevated Na content and high concentrations of potentially toxic metals [7,8] and potential leaching (groundwater, surface waters, soils, and ocean ecosystems), it can pose a significant environmental hazard if not stored properly. Its storage is a critical environmental problem [9,10,11,12,13,14,15,16]. This material is typically stored in dams, which demands prior preparation of the disposal area and includes monitoring and maintenance during the storage period [17].

A small residual amount of the sodium hydroxide used in the process remains with the residue. Various stages in the solid/liquid separation process have been introduced to recycle as much hydroxide as possible from the residue back into the Bayer process to make the process as efficient as possible and to reduce production costs. These modifications also decrease the final alkalinity of the residue, making it easier and safer to handle and store [18].

The present study discusses the technical viability of RM valorization. The authors proposed the utilization of residual WAS from the aluminum industry for the first time as a catalyst in the transesterification reaction in aqueous medium for the production of fatty acid methyl esters (FAME).

FAME, or biodiesel, according to the American Society for Testing and Materials (ASTM) and European standards (EN), is a fuel consisting of “long chain fatty acids of mono-alkyl esters derived from renewable fatty raw material such as animal fats or vegetable oils” [19]. FAME content needs to be higher than 96.5 wt.%. The total glycerol, including bound glycerol [in glycerides such as monoglycerides (MGs), diglycerides (DGs), and triglycerides (TGs)] and unbound glycerol (free glycerol), needs to be limited to 0.24 and 0.25 wt.% by ASTM and EN standards, respectively.

In the search for an environmentally friendly method for biodiesel synthesis by transesterification of TGs, several alternatives for catalysis have been explored. In general, catalysts that can be used for producing biodiesel are divided into three categories: acidic, alkaline, and biocatalysts. Acidic and alkaline catalysts are classified into two groups: homogeneous and heterogeneous catalysts.

Catalysts play a vital role in the transesterification process. Both the amount and type of catalyst affect the rate of reaction and conversion efficiency. Homogeneous catalysts function in the same phase as the reactants and can be categorized into homogeneous base catalysts and homogeneous acid catalysts. Currently, most FAMEs are produced by the base-catalyzed transesterification reaction because of its high conversion rate, negligible side reactions, and short reaction time. It is a low-pressure and low-temperature process, which occurs without the formation of intermediate substances. Despite these advantages, homogeneous base catalysts have some weaknesses. The production of biodiesel from feedstocks with a high free fatty acid (FFA) content is limited. It was reported by some researchers that homogeneous base catalysts are only effective for the production of FAME via the transesterification process using the feedstocks with an FFA content of less than 2 wt.% [20]. When FFA content is >2%, the catalyst reacts with FFA to produce soap and water. The soap inhibits the separation of FAME and glycerin, and the water can hydrolyze the esters in a reaction that competes with the transesterification.

In transesterification reactions of TGs catalyzed by homogeneous bases, FAME and glycerol are produced. Zhang, Stanciulescu, and Ikura (2009) demonstrated that the use of phase transfer agents (PTA) greatly increased the rate of the base-catalyzed transesterification reaction [21]. A product containing 96.5 wt.% was obtained after only 15 min of rapid reaction at 60 °C in the presence of tetrabutylammonium hydroxide or acetate. The reaction was performed in the presence of MeOH, glycerol, refined and bleached soybean oil, and the basic catalyst, without the presence of water.

Recently, we also studied the acid-catalyzed transesterification reaction using WCO and a solid acidic catalyst (SiO_2_-SO_3_H) in the presence of quaternary ammonium salts as co-catalysts in toluene and DMSO [22]. We decided to study the transesterification reaction in a medium containing WAS. This study sought to utilize the industrial residues (RM; WAS) to increase their value and decrease the environmental problems resulting from the storage of these residues.

## 2. Experimental

### 2.1. Raw Materials and Chemicals

RM containing WAS was collected from ALCOA (Juruti, PA, Brazil). WCO (soybean) was donated by the university restaurant, and it was filtered through silica gel, which removed any fatty acids (FA) and polar and polymeric substances prior to use. The physical parameters determined for the yellow oil were the viscosity (41.2 mPa) and the density (0.883 g·mL^−1^). MeOH (analytical grade) was supplied by Vetec, São Paulo, Brazil.

### 2.2. Typical Procedures

#### 2.2.1. Standardization of the WAS Generated during the Processing of Bauxite into Alumina Using the Bayer Process

WAS (1.60 mL) was diluted to 100 mL. An aliquot of 1.00 mL was removed and diluted again to 100 mL. A 25 mL aliquot was collected from this solution and titrated with a 0.0017 M HCl solution using 0.48 mL of HCl solution. Titration was accomplished using an SI Analytics Titrator TitroLine^®^ 7000 potentiometric titrator. The concentration of WAS was determined as 0.204 mol L^−1^.

#### 2.2.2. Elemental Analysis Based on Energy-Dispersive X-ray Fluorescence (EDXRF) of Dehydrated WAS (Dry Solid) (Chemical or Elemental Composition of WAS)

The pH of the original WAS from RM was 13.3. The WAS sample was previously dried for 24 h in a muffle furnace at 150 °C. The equipment used in this procedure was an X-ray fluorescence spectrometer, model EDX 720 (Shimadzu^®^; Kyoto, Japan), equipped with an X-ray tube and the use of liquid nitrogen for cooling. The software used was PCEDX, version 1.11 Shimadzu^®^. Energy scattering X-ray fluorescence is a non-destructive, multi-element analytical technique capable of identifying elements with an atomic number Z greater than or equal to 12. When the electrons of the innermost layer of the atom (for example, K and L) interact with photons in the X-ray region, then these electrons are ejected, creating a vacancy. To promote stability, the electronic vacancies are immediately filled by electrons from the closest layers (Kα, Kβ or Lα, Lβ), resulting in an excess of energy in the process, which is manifested in the form of the emission of X-rays characteristic of each atom present in the sample.

The EDXRF is an apparatus used for the quantitative and quali-quantitative determination of chemical elements in a wide range of samples. The analyses carried out in this work used qualitative and quantitative determination, with only the pre-calibration of the equipment (Al ≥ 80% and the detection of Sn and Cu), using only atmospheric air, and restricting the detection of metals included between _13_Al and _92_U. The samples were placed in sample holders made of polypropylene film and the analysis conditions were as follows: 10 mm collimator, scans with voltages of 0–40 KeV (Ti-U) and 0–20 KeV (Na-Sc) with a time of 100 s for each sample.

In this method, the material to be analyzed is targeted with an X-ray beam that interacts with the atoms of the sample and causes the ionization of the innermost layers of the atoms. The filling of the resulting vacancies by more peripheral electrons induces the emission of X-rays characteristic of the constituent elements of the sample. The elemental composition of WAS analyzed by EDXRF was the following (in wt.%) (see Table 1): Al, 54.58; Si, 39.54; K, 1.99; V, 1.88; Ga, 0.95; Cs, 0.33; Cr, 0.174; Fe, 0.142; Br, 0.089; Cu, 0.079; Mo, 0.069; Tl, 0.064; Ag, 0.064; and Zr, 0.047. EDXRF was used as a method for semi-quantitative elemental analysis, and the titanium derivatives were not quantified. Their concentrations were lower than the quantification limit. However, their presence was clearly observed in the powder XRD spectra.

#### 2.2.3. Experimental X-ray Diffraction of Dehydrated WAS and Characterization of the Material by Powder XRD

The measurements were performed with a Shimadzu model XRD-6000 diffractometer using CuKα monochromatic radiation (λ = 0.15406 nm–40 kV and 30 mA) at a scan rate of 2.0 degrees·s^−1^, covering the 2θ scale from 10–80°. WAS was dried at 150 °C before experimental X-ray diffraction. The XRD patterns of the dehydrated WAS (200 °C) are shown in Figure 1. The most common mineral phases present in the WAS were hematite (Fe_2_O_3_), goethite (FeOOH), gibbsite (Al(OH)_3_), a boehmite (AlOOH), titanium mineral as anatase (Tetragonal–TiO_2_), rutile (Ditetragonal dipyramidal–TiO_2_), and the hydrogarnets group [Ca_3_Al_2_(SiO_4_)_3−x_(OH)_4x_], such as Fe_3_Al_2_(SiO_4_)_3_, quartz (SiO_2_), and perovskite (CaTiO_3_). Iron crystalline phases were identified in the XRD diffractograms as hematite and goethite phases, and they were attributed in 2θ = 34.4°, 35.5°, 41.5°, 50.6°, 53.7°, 56.5°, 23.4°, and 33.5°, respectively. Aluminum crystalline phases were identified as gibbsite and boehmite and were attributed in 2θ = 18.6°, 21.1°, 30.2°, 38.1°, and 48.3°, respectively. The aluminosilicate crystalline phase was attributed to the hydrogarnets group with 2θ = 40°, 44.8°, 52°, 54.8°, and 67.4°. The silicon crystalline phase was attributed to quartz with 2θ = 60.8° and 63.9°. Finally, titanium crystalline phases were the last type of material identified as titanium dioxide and perovskite by XRD diffractograms, with signals in 2θ = 26.1°, 27.8°, 62.1°, 46.7°, 59°, and 69.3°, respectively. The WAS composition agrees with that of the EDXRF identification, in which aluminosilicate was confirmed to be the principal component, with traces of iron, chromium, and copper that can act as active species in the catalysis or in a synergistic catalysis process with potassium [23,24].

#### 2.2.4. Thermal Gravimetric (TG) Analysis Technology of Dehydrated WAS

WAS had been dried at 150 °C before thermogravimetric analysis. The TG curve (Figure 2, red) of the WAS indicated an intense mass loss (i.e., 76%) between the initial heating phase, 30 °C up to 253 °C, with a corresponding endotherm displayed by the simultaneous DTA curve (Figure 2, black), which is equivalent to the removal of moisture from the sample. A small endothermic peak around 300 °C (T_onset_ 274.4 °C) was observed in the DTA curve without mass loss in the same temperature range. It corresponds to the melting of the residue obtained at a temperature greater than 253 °C. The highlighted photos show the crucible containing the sample before (upper left) and after (upper right) the analysis. Note the presence of a white solid residue representing 24% of the sample.

#### 2.2.5. Reacting the TGs from WCO with MeOH Using WAS as Catalyst

The procedure utilized for the transesterification reaction was based on various trials to determine the optimum conditions for this reaction. A 150-mL round bottom flask, equipped with a reflux condenser, containing WAS (0.204 mol L^−1^; or 0.019% *w*/*w* of WCO) with MeOH (3.75 mL, 2.9738 g, 0.0928 mol; or a 1:18 molar ratio of WCO/MeOH) and WCO (4.4170 g, 5.0 mL; 5.0510^−3^ mol) were mixed, and the mixture was refluxed for 30 min at 60 °C. The mixture was cooled and transferred to a separatory funnel where the biofuel-containing upper phase was separated from the lower phase containing glycerol by decantation. The MeOH was removed from the biodiesel phase on a rotary evaporator, purified by distillation, and used in new reaction processes within this study. The recovered glycerol was stored for future treatments. The biofuel phase was dissolved in hexane (20 mL), extracted with 20 mL of a saturated solution of NaCl, dried over MgSO_4_, and concentrated.

### 2.3. WCO and Biodiesel Analysis

The official methods proposed by ISO 12966 were used to determine the compositional profile by gas chromatography using a flame ionization detector (GC-FID) (Shimadzu GC-2010). The chromatographic system used to separate and identify FFAs (wt.%) included a cross-bound polyethylene glycol capillary column (Supelco SP 2560, 100 m × 0.25 mm × 20 µm). The initial temperature was 60 °C for 2 min; the temperature increased to 220 °C at 10 °C·min^−1^, and finally, to 240 °C at 5 °C·min^−1^, where it was held for 7 min. The injector and detector temperatures were 350 °C, and the sample (0.5 µL injected) was dissolved in 99% isooctane.

The EN 14103 and the Brazilian Technical Standards Association (ABNT NBR 15908) were used to quantify FAMEs and remaining mono-, di-, and triglycerides (MG, DG, TG) in the FAME (biodiesel). For quantification of FAMEs, a Thermo Trace GC-Ultra chromatograph equipped with a flame ionization detector and a Thermo Scientific TR-BD (FAME) Capillary GC Column (L × I.D. 30 m × 0.25 mm, df 0.25 μm) containing a polyethylene glycol stationary phase was used, according to the EN 14103 analytical procedure. Pure methyl nonadecanoate (C19:0, Sigma-Aldrich—Sao Paulo, Brazil) was used as an internal standard to normalize the peak areas of the chromatograms. The integration was achieved from the methyl hexanoate (C6:0) peak to that of the methyl nervonate (C24:1), including all the peaks identified as FAMEs. To analyze the FAME samples, approximately 100 mg (accuracy ± 0.1 mg) of homogenized sample and approximately 100 mg (accuracy ± 0.1 mg) of nonadecanoic acid methyl ester were weighed in a 10 mL vial and diluted with 10 mL of toluene before injection into the equipment. All the samples were prepared in duplicate. Chromatographic conditions are described as follows: (a) column temperature: 60 °C held for 2 min, programmed at 10 °C·min^−1^ to 200 °C, and then programmed at 5 °C·min^−1^ to 240 °C; the final temperature was held for 7 min; (b) injector and detector temperature: 250 °C; (c) helium carrier gas flow rate: 1–2 mL·min^−1^; a minimum flow rate of 1 mL·min^−1^ was warranted when operating at the maximum temperature; (d) injected volume: 1 μL; and (e) split flow: 100 mL·min^−1^.

For the quantification of the glycerides (MG, DG, and TG), a Shimadzu GC2010 chromatograph equipped with a flame ionization detector was used according to the ASTM D6584 analytical procedure. The chromatographic system was configured to separate and identify MG, DG, and TG with a CrossbondTM 5% Phenyl/95% dimethylpolysiloxane capillary column (Zebron ZB-5HT, 30 m × 0.32 mm × 0.1 mm–Phenomenex, Torrence, CA, USA) with on-column injection. The initial temperature in the capillary column was 50 °C (1 min); the temperature increased to 180 °C at 15 °C min^−^ to 230 °C at 7 °C·min^−1^, and finally, to 380 °C at 20 °C·min^−1^, where it was held for 10 min. The injector and detector temperatures were 380 °C, and the sample (0.5 mL injected) was prepared using heptane 99%. ^1^H- and ^13^C-NMR spectra were recorded on Bruker *Avance* 400 and *Avance* 500 spectrometers. These data are included in the Appendix A.

#### Thermogravimetry Analysis

The thermal behavior of FAME was evaluated by thermogravimetry (TG), and the data were treated simultaneously in the first derivative, thermogravimetry derivative (DTG), and differential thermal analysis (DTA). TG/DTA curves were obtained on a DTG60H Shimadzu thermobalance; the heating rate was 10 °C·min^−1^ and the temperature range was 25–350 °C under a controlled nitrogen atmosphere at 50 mL·min^−1^ and under oxidizing conditions (synthetic air). An open alumina crucible had an accurately weighed sample mass of about 20 mg. The derivative curve was obtained by T.A. data software.

## 3. Results and Discussion

The compositional profile analysis of the WCO used in this work is described in Table 2. The main fatty acids (FAs) in that WCO were linoleic (C18:2) and oleic (C18:1) acids; accordingly, the mean molecular weight (MW) of the FAs was determined to be 277.41 g·mol^−1^, and the mean molecular mass (MM) of the TGs was 873.22 g·mol^−1^. The composition of the WCO was very similar to that of soybean oil (SO) described in the literature [25]. The following profile was considered for calculating the molar ratio of WCO: MeOH for the transesterification reaction.

Lit [17]: palmitic acid, 11.6%; stearic acid, 3.22%; oleic acid, 25.09%; linoleic acid, 52.93%; linolenic acid, 5.95%; others, 1.08%.

Sodium methoxide (MeONa) is formed by the deprotonation of MeOH and is frequently prepared by treating MeOH with metallic sodium [26]:Na + CH_3_OH → CH_3_ONa + 1/2H_2_(1)

The resulting solution, MeONa/MeOH, which is colorless, is often used as a source of MeONa. The MeONa absorbs CO_2_ from the air to form MeOH and Na_2_CO_3_, thereby diminishing the alkalinity of the base.
CH_3_ONa + CO_2_ + H_2_O → 2 CH_3_OH + Na_2_CO_3_(2)

MeONa is highly basic and reacts with water, resulting in MeOH and NaOH. In this work, we used a solution of WAS (0.204 mol/L, determined by potentiometric titration) which leads to a very low concentration of MeONa. Thus, we believe that the reaction did not occur as a result of a nucleophilic attack by MeONa on the ester carbonyl groups of TGs, DGs, and MGs.

In an unprecedented process, biodiesel was produced by catalysis using WAS obtained from the aluminum industry for the transesterification reaction of WCO. The WAS-catalyzed transesterification is a rapid reaction at both initial and final reaction stages because it is not limited by mass transfer between the polar water/MeOH/glycerol phase and the non-polar MGs, DGs, and TGs phase.

Zhang, Stanciulescu, and Ikura (2009) used a total OH^−^/oil molar ratio of 0.22 [27]. In this study, we believe that the low molar concentration of hydroxide (0.019% *w*/*w* of WCO or 0.0040 mol/mol of WCO) is related to gibbsite (Al(OH)_3_) and to the fact that the hematite (Fe_2_O_3_) contained in WAS forms an anionic iron species in the presence of hydroxide ion. According to Ishikawa, Yoshioka, Sato, and Okuwaki (1997), hematite in the presence of sodium hydroxide leads to the formation of NaFeO_2_ (FeO_2_^−^) [28].
Fe_2_O_3_(s) + 2NaOH(aq) → 2FeO_2_^−^ + 2Na^+^ + H_2_O(3)

Species like FeO_2_^−^ might facilitate the formation of the nucleophilic species MeO^−^ in a medium containing MeOH. Wang et al. (2022) have shown that the NaFeO_2_-Fe_3_O_4_ composite from blast furnace dust acts as an efficient catalyst for the production of biodiesel [29]. They obtained the catalyst by treating the blast furnace dust with different proportions of sodium carbonate and calcining the mixture. The catalyst was recycled, and high yields were still obtained (Na_2_CO_3_⋅H_2_O@BFD300): 93.00 wt.% at the eighth use (NaHCO_3_@BFD300) and 96.16 wt.% at the seventh use (Na_2_CO_3_⋅10H_2_O@BFDun). The highest yields for sustainable biodiesel production were obtained with Na_2_CO_3_⋅H_2_O@BFD300 catalyst as a result of the reaction of impregnated Na_2_CO_3_ with Fe_2_O_3_ in the blast furnace dust to produce stable and active nanocomponents of NaFeO_2_ (32.42 nm) and a magnetic nanocomponent of Fe_3_O_4_ (3.14 nm and Ms of 6.16 Am_2_/kg). Blast furnace dust was a suitable raw material for catalyst synthesis to produce soybean biodiesel by the transesterification reaction in a non-aqueous medium. This composite would also be formed in the residue from aluminum production and would explain the high yield of biodiesel obtained, even though the free hydroxide concentration was very low.

Liu et al. [30] and Wang et al. [31] reported that the alkaline content of the material from the Bayer process includes soluble NaOH, Na_2_CO_3_, NaAlO_2_, Na_2_SiO_3_, and sodium aluminosilicate hydrate (Na_2_O·Al_2_O_3_·xSiO_2_·mH_2_O; zeolite) generated from the reaction between the bauxite and the highly alkaline solution. Pampararo and Debecker [32] observed that sodium aluminate (NaAlO_2_) was an effective catalyst for the preparation of biodiesel from refined sunflower oil in a moisture-free process. The catalyst was calcined to remove any moisture because the material was hygroscopic. The oil was also heated to remove any moisture. They showed that the activity of the catalyst was a function of its basicity. The waste material utilized in our study contained 54.581% aluminum [gibbsite (Al(OH)_3_), a boehmite (AlOOH), and hydrogarnets Ca_3_Al_2_(SiO_4_)_3−x_(OH)_4x_], and one would expect that this component would be the principal agent responsible for the catalysis of the transesterification reaction, either directly or indirectly via the equilibrium with the hematite (Fe_2_O_3_), goethite (FeOOH), anatase (tetragonal–TiO_2_), rutile (ditetragonal dipyramidal–TiO_2_), and perovskite (CaTiO_3_). The solution was used directly in the form that it was produced by in the industry. No calcination was performed. The activity of WAS was slightly higher than that of the NaFeO_2_-Fe_3_O_4_, Na_2_CO_3_⋅H_2_O@BFD300, and Na_2_CO_3_⋅10H_2_O@BFDun described above.

### 3.1. Transesterification of TGs from WCO with MeOH Using WAS

In this study, an excess of MeOH was mixed with standardized WAS and WCO and refluxed at 60 °C. The progress of the reaction was monitored using thin layer chromatography (TLC). The total consumption of TGs occurred after 60 min. The FAMEs and glycerides (MG, DG, and TG) contained in the biodiesel phase were confirmed by GC-FID using the methods defined in EN 14103 and ASTM D6584. The composition of the products (FAMEs and glycerides) is presented in Table 2; they represent the average values of five different measurements. More than 30 experiments were performed in which the proportion of WAS catalyst to WCO was varied. The greatest efficiency was observed when the mass ratio of WAS to WCO was 1:4700.

The composition of the products (FAMEs and glycerides) after 10.0 min of reaction is presented in Table 3; they represent the average values of five different measurements. The yields for completed transesterification reactions are presented in Table 4.

A comparison of the results obtained using the WAS catalyst and a traditional catalyst (KOH) for FAME (biodiesel) production from WCO as feedstock is presented in Table 5. Refaat et al. [33] stated that FAME production is worthy of continued study and optimization of production procedures because of its environmentally beneficial attributes and its renewable nature. Their study was intended to consider aspects related to the feasibility of the production of biodiesel from WCO in an attempt to help reduce the cost of biodiesel and reduce waste and pollution resulting from WCO. The variables affecting the yield and characteristics of the biodiesel produced from WCO were studied. The best yield was obtained using a 6:1 MeOH/WCO with KOH as the catalyst (1%) at 65 °C for 60 min. The yield obtained from WCO reached 96.15% under optimum conditions.

Xiangmei Meng, Guanyi Chen, and Yonghong Wang [34] used WCO, which contained large amounts of free fatty acids produced in restaurants and was collected by the Environmental Protection Agency in the main cities of China. The optimum experimental conditions, which were obtained from the orthogonal test, were MeOH/WCO molar ratio 9:1 with 1.0 wt.% NaOH, a temperature of 50 °C, and 90 min. The 6:1 MeOH:WCO molar ratio was most suitable in the process, and WCO conversion efficiency was 89.8%.

### 3.2. Determination of Thermophysical Properties of FAME

The significant properties of FAME are determined by the various tests and methods as per the ASTM specifications. Table 6 gives the standard test methods used for the determination of various properties of FAME.

### 3.3. Thermogravimetry Analysis of FAME

Under inert conditions (N_2_ atmosphere), FAME ignites at 140 °C, as is seen in the TG curve (top Figure 3, red), involving two overlapping thermal decomposition mechanisms illustrated by the dotted highlight of the DTG curve (top Figure 3, pink). In the thermal decomposition range, 140–295 °C, a broad endothermic event characteristic of this phenomenon occurs, as is seen in the DTA curve (top Figure 3, black). The biodiesel was almost completely consumed, with 98.6% loss in mass. The photos in the detail of Figure 3, top, show the alumina crucible containing the sample before ignition and after firing up to 350 °C.

Biodiesel underwent ignition from 124 °C under oxidizing conditions (under an atmosphere of synthetic air), as observed in the TG curve (Figure 1, bottom, red); that is, at a lower temperature than that observed under inert conditions because the supply of oxygen favors the combustion process. Some inflections of the TG curve occur, as is seen in dotted details of the DTG curve (Figure 3, bottom, pink), showing the complexity of the mechanisms of thermal decomposition of biodiesel under oxidizing conditions. This phenomenon is also marked by a broad endothermic curve (DTA curve, Figure 3, bottom, black) over the full range of decomposition (124–307 °C). Biodiesel burned almost completely, with 98.6% mass loss. The photos in the detail of Figure 3, top, show the alumina crucible containing the sample before ignition and after firing up to 350 °C. Note the intensity of carbonization of the sample, consistent with the oxidizing condition.

## 4. Conclusions

The transesterification reaction of TGs from WCO was catalyzed by WAS {containing hematite (Fe_2_O_3_), goethite (FeOOH), gibbsite (Al(OH)_3_), a boehmite (AlOOH), anatase (Tetragonal–TiO_2_), rutile (Ditetragonal dipyramidal–TiO_2_), hydrogarnets [Ca_3_Al_2_(SiO_4_)_3−x_(OH)_4x_], SiO_2_ (quartz), and perovskite (CaTiO_3_)}. The WAS (catalytic red mud solution) was used successfully for the transesterification of TGs from WCO with MeOH. Analysis of the quantities of TGs, DGs, MGs, and FAMEs in the products indicated excellent catalytic behavior of the WAS. The WAS-catalyzed transesterification rate was indicated by the high FAME yield obtained after only 30 min of reaction. The rapid transesterification observed can be explained by the fact that WAS can facilitate ion transfer between the polar water/MeOH/glycerol phase and non-polar WCO phase, overcome mass transfer limitations, and speed up reaction rates. Product analyses showed that a FAME content greater than 99.5 wt.% was achieved after only 30 min of rapid transesterification. Free and total glycerol contents in the final products after 30 min of transesterification were lower than the maximum legal limits in standard specifications for FAME. The catalyst was suitable for the synthesis of FAME from WCO by the transesterification reaction in aqueous medium. This method employing WAS as a catalyst could enable recycling of the waste alkaline solution from the bauxite process, minimize contaminants, and reduce the cost of the catalyst. In addition, it would be of great environmental value through the decontamination of groundwater and soils, as well as the elimination of areas intended for the disposal of these alkaline solutions that result from the Bayer process in the production of alumina. The use of this industrial residue as a catalyst by biodiesel-producing industries could lead, in the short term, to a total replacement of traditional catalysts by the use of WAS and zero investment in traditionally used catalysts. This highly efficient and low-cost WAS catalyst could make the process of FAME production more economical. The global sodium methoxide solution as a biodiesel catalyst market is estimated to value at around US $0.3 bn in 2021 and is expected to register a CAGR of 3.1% [37]. In addition to FAME production, such environmentally friendly WAS catalysts should find application in a wide range of other important base-catalyzed organic reactions, such as the Michael reaction or Michael 1,4-addition.

## Figures and Tables

**Figure 1 bioengineering-10-00692-f001:**
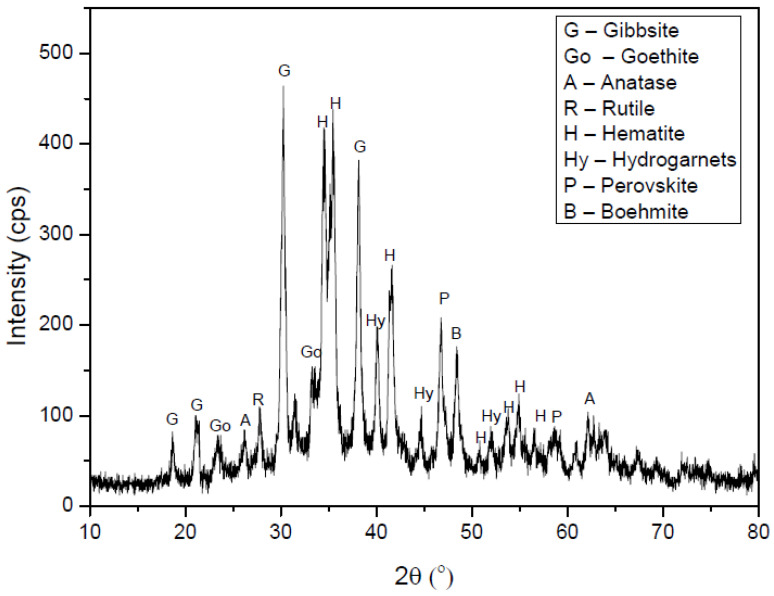
PXRD (X-ray diffraction patterns) of dehydrated WAS (dry solid).

**Figure 2 bioengineering-10-00692-f002:**
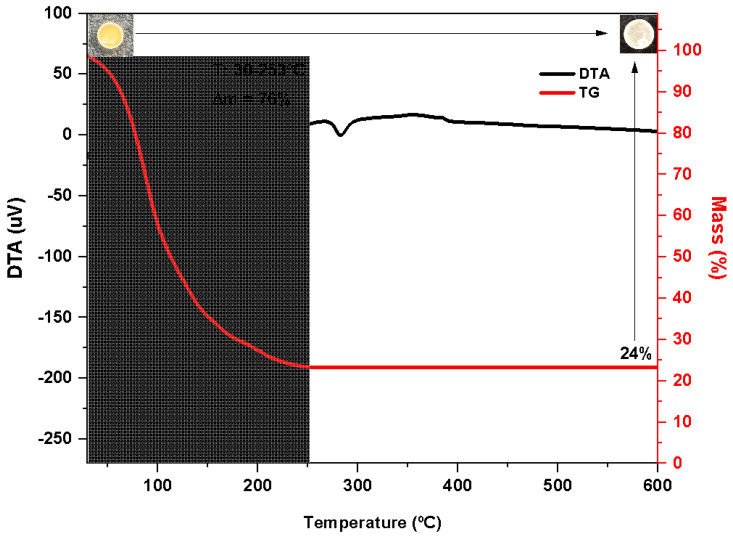
Thermal analysis WAS.

**Figure 3 bioengineering-10-00692-f003:**
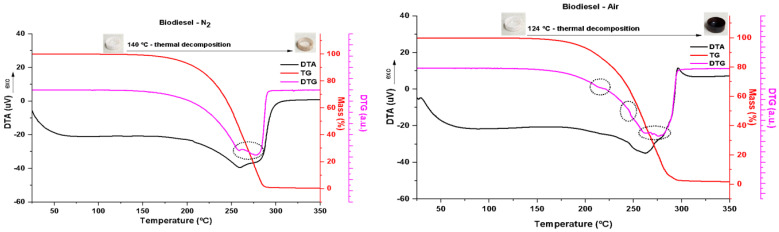
Thermal behavior of biodiesel under inert (N_2_) conditions (**left graph**) and under oxidant (air) conditions (**right graph**).

**Table 1 bioengineering-10-00692-t001:** The elemental composition of WAS determined by EDXRF.

Elemental Composition	wt.%
Al	54.58
Si	39.54
K	1.99
V	1.88
Cs	0.330
Cr	0.174
Fe	0.142
Br	0.089
Cu	0.079
Mo	0.069
Tl	0.064
Ag	0.064
Zr	0.047

**Table 2 bioengineering-10-00692-t002:** FAs composition of the WCO used in this study.

FA	MW (g·mol^−1^)	wt.%
Linoleic acid (C18:2)	280.45	51.66
Oleic acid (C18:1)	282.46	26.52
Palmitic acid (C16:0)	256.43	10.41
Linolenic acid (C18:3)	278.43	5.55
Stearic acid (C18:0)	284.48	3.91
Others	-	1.95
Average MW of FAs (g·mol^−1^)	277.41	
MM of TGs (g·mol^−1^)	873.22	

**Table 3 bioengineering-10-00692-t003:** Composition of the WCO feedstock and the product mixture using refluxing MeOH.

Products	WCO (wt.%)	WAS Catalyst Mixture *
Triacylglycerides (%)	96.2 ± 2.84	5.9 ± 0.17
Diacylglycerides (%)	2.80 ± 1.80	11.0 ± 0.32
Monoacylglycerides (%)	1.0 ± 0.92	22.9 ± 0.68
FAME (%)	0	60.2 ± 1.78

* The composition of the products (FAMEs and glycerides) after 10.0 min of reaction.

**Table 4 bioengineering-10-00692-t004:** Composition of the product of transesterification of TGs from WCO with MeOH using WAS catalyst.

	WAS Catalyst Mixture
Assay	Yield * (% *w*/*w*)
FAME	99.5 ± 2.94
Free glycerol (FG)	0.01 ± 0.01
Total glycerol (TG)	0.07 ± 0.03
MG	0.26 ± 0.24
DG	0.06 ± 0.02
TG	0.01 ± 0.01

* The composition of the products (FAMEs and glycerides) after 30.0 min of reaction at 60 °C.

**Table 5 bioengineering-10-00692-t005:** Comparison of the yield of the transesterification with WAS catalyst and traditional catalysts (NaOH and KOH) for biodiesel production using WCO as feedstock.

	WAS Catalyst	NaOH Catalyst	KOH Catalyst
MeOH/WCO molar ratio	18:1	9:1	6:1
Temperature °C	60	50	65
Time (min)	30	90	60
FAME (%)	99.5%	89.8%	96.15%

**Table 6 bioengineering-10-00692-t006:** ASTM standards for FAME properties and the experimental values.

Property	FAME	Reference	Test Standard
Kinematic Viscosity (40 °C; mm^2^/s)	5.03	[35]	ASTM D 445-04e
Density	0.87	[35]	Density ASTM D7371-12
Cloud Point	−1	[35]	Cloud Point ASTM-D 2500-05
High heating value	41.28	[35]	ASTM D-240-02
Cloud Filter Plugging Point	−7	[35]	ASTM D6377-05
Cetane Number	61	[35]	ASTM D 613-05
Pour Point	−16	[36]	ASTM-D97
Flash Point	164	[36]	EN ISO 2719

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
