# Peer review of "Environmentally Friendly New Catalyst Using Waste Alkaline Solution from Aluminum Production for the Synthesis of Biodiesel in Aqueous Medium"

_bioengineering, 2023, doi:10.3390/bioengineering10060692_

Round 1

Reviewer 1 Report

Report on the manuscript bioengineering-2408309; entitled “Environmentally friendly new catalyst using waste alkaline solution from aluminum production for the synthesis of biodiesel in aqueous medium”.

The submitted manuscript [bioengineering-2408309] should be revised. The following points should be addressed:

1. The submitted manuscript should be revised to be free from editing or grammar errors.

2. In abstract “soluble hematite”, hematite is enough because “soluble” will open many questions and issues.

3. In XRD analysis, the values of 2θ should be clearly mentioned in the discussion.

4. In TGA analysis, there is a small endothermic peak around 300 C in only DTA and no change in TGA. Please, explain it.

5.  In line 53, the reference [21] hasn’t the mentioned equation so, it was cited incorrectly. Please, revise it!

6. Could the authors provide EDX analysis of dehydrated WAS (dry solid) to confirm its chemical content beside XRD results.

Major revision in language terms

Author Response

Comment: 1. The submitted manuscript should be revised to be free from editing or grammar errors.

Response: The manuscript was revised, and any error encountered were corrected.

Comment: 2. In abstract “soluble hematite”, hematite is enough because “soluble” will open many questions and issues.

Response: the adjective soluble was removed.

Comment: 3. In XRD analysis, the values of 2θ should be clearly mentioned in the discussion.

Response: The values of 2θ were clearly mentioned in the XRD analysis.

Comment: 4. In TGA analysis, there is a small endothermic peak around 300 C in only DTA and no change in TGA. Please, explain it.

Response: The small endothermic peak around 300 C in only DTA with no change in TGA was explained. It represents the melting of the residue obtained at a temperature greater than 253 ºC. 

Comment: 5.  In line 53, the reference [21] hasn’t the mentioned equation so, it was cited incorrectly. Please, revise it!

Response: The citation of the reference for the equation was corrected.

Comment: 6. Could the authors provide EDX analysis of dehydrated WAS (dry solid) to confirm its chemical content beside XRD results.

Response: The analyses were performed with the dehydrated sample, as described in the experimental section.

Reviewer 2 Report

In this manuscript, authors have used an environmentally friendly catalyst for biodiesel production using WCO. The new catalyst was obtained from waste alkaline solution from aluminum company. The manuscript has certainly a novelty factor, below are the comments to further improve the manuscript:

·       Provide composition of WAS in a table

·       I don’t see optimization of catalyst weight % for biodiesel production. Its something which need to be considered

·       I think authors forgot to provide SD in table 2

·       In table 3, SD for some of the components is more than actual value of component. Please correct the table

·       Provide table comparing the new catalyst and traditional catalysts (FAME yield and efficiency) for biodiesel production using WCO as feedstock

·       For biodiesel produced using WAS, compare the properties with ASTM standards or any other standard

·       Just a general query: How much impact a catalyst has on biodiesel production cost? This will provide cost-effectiveness of new catalyst

·       In conclusions section, provide future research directions based on this study: like techno-economic assessment, environmental assessment etc.

Author Response

  • Comment: Provide composition of WAS in a table

Response: The composition of WAS was included in Table 1.

Comment:·       I don’t see optimization of catalyst weight % for biodiesel production. Its something which need to be considered

Response: The determination of the optimum catalyst weight was mentioned and the weight was cited.

  • Comment:  I think authors forgot to provide SD in table 2

Response: The SDs were included in Table 2

  • Comment:   In table 3, SD for some of the components is more than actual value of component. Please correct the table

Response: was correct.

  • Comment:     Provide table comparing the new catalyst and traditional catalysts (FAME yield and efficiency) for biodiesel production using WCO as feedstock

Response: Table 5 was included in which the efficiency of the WAS catalyst was compared with those of traditional KOH and NaOH catalysts.

  • Comment:For biodiesel produced using WAS, compare the properties with ASTM standards or any other standard

Response: Table 6, in which the Standard values for biodiesel are listed, was added.

  • Comment:     Just a general query: How much impact a catalyst has on biodiesel production cost? This will provide cost-effectiveness of new catalyst

Response: The cost of a traditional catalyst was added to the Conclusion. The cost of the WAS catalyst is zero.

  • Comment:   In conclusions section, provide future research directions based on this study: like techno-economic assessment, environmental assessment etc.

Response: A general comment regarding additional possible uses of the catalyst and environmental advantages of its use was included in the Conclusion.

Round 2

Reviewer 1 Report

Accepted

It's Ok

Author Response

Thank you for correction!

Reviewer 2 Report

Please include results in Table 6, if authors have tested FAME properties.

Author Response

Data referring to FAME were added in Table 6.